# LEARNING CONSISTENT DEEP GENERATIVE MODELS FROM SPARSE DATA VIA PREDICTION CONSTRAINTS

## ABSTRACT

We develop a new framework for learning variational autoencoders and other deep generative models that balances generative and discriminative goals. Our framework optimizes model parameters to maximize a variational lower bound on the likelihood of observed data, subject to a task-specific prediction constraint that prevents model misspecification from leading to inaccurate predictions. We further enforce a consistency constraint, derived naturally from the generative model, that requires predictions on reconstructed data to match those on the original data. We show that these two contributions – prediction constraints and consistency constraints – lead to promising image classification performance, especially in the semi-supervised scenario where category labels are sparse but unlabeled data is plentiful. Our approach enables advances in generative modeling to directly boost semi-supervised classification performance, an ability we demonstrate by augmenting deep generative models with latent variables capturing spatial transformations.

## 1 INTRODUCTION

We develop broadly applicable methods for learning flexible models of high-dimensional data, like images, that are paired with (discrete or continuous) labels. We are particularly interested in *semi-supervised* learning (Zhu, 2005; Oliver et al., 2018) from data that is sparsely labeled, a common situation in practice due to the cost or privacy concerns associated with data annotation. Given a large and sparsely labeled dataset, we seek a single probabilistic model that *simultaneously* makes good predictions of labels and provides a high-quality generative model of the high-dimensional input data. Strong generative models are valuable because they can allow incorporation of domain knowledge, can address partially missing or corrupted data, and can be visualized to improve interpretability.

Prior approaches for the semi-supervised learning of deep generative models include methods based on *variational autoencoders* (VAEs) (Kingma et al., 2014; Siddharth et al., 2017), *generative adversarial networks* (GANs) (Dumoulin et al., 2017; Kumar et al., 2017), and hybrids of the two (Larsen et al., 2016; de Bem et al., 2018; Zhang et al., 2019). While these all allow sampling of data, a major shortcoming of these approaches is that they do not adequately use labels to inform the generative model. Furthermore, GAN-based approaches lack the ability to evaluate the learned probability density function, which can be important for tasks such as model selection and anomaly detection.

This paper develops a framework for training *prediction constrained variational autoencoders* (PC-VAEs) that minimize application-motivated loss functions in the prediction of labels, while simultaneously learning high-quality generative models of the raw data. Our approach is inspired by the prediction-constrained framework recently proposed for learning supervised topic models of "bag of words" count data (Hughes et al., 2018), but differs in four major ways. First, we develop scalable algorithms for learning a much larger and richer family of deep generative models. Second, we capture uncertainty in latent variables rather than simply using point estimates. Third, we allow more flexible specification of loss functions. Finally, we show that the generative model structure leads to a natural consistency constraint vital for semi-supervised learning from very sparse labels.

Our experiments demonstrate that *consistent prediction-constrained* (CPC) VAE training leads to prediction performance competitive with state-of-the-art discriminative methods on fully-labeled datasets, and *excels* over these baselines when given semi-supervised datasets where labels are rare.

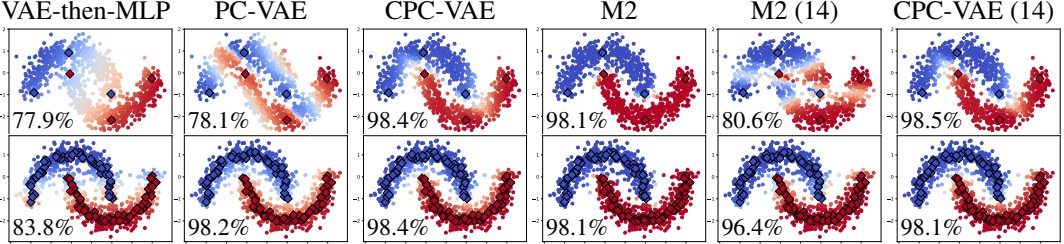

Figure 1: Predictions from SSL VAE methods on half-moon binary classification task, with accuracy in lower corner. Each dot indicates a 2-dim. feature vector, colored by predicted binary label. *Top:* 6 labeled examples (diamond markers), 994 unlabeled. *Bottom:* 100 labeled, 900 unlabeled. First 4 columns use $C = 2$ encoding dimensions, last 2 use $C = 14$. M2 (Kingma et al., 2014) classification accuracy *deteriorates* when increasing model capacity from 2 to 14, especially with only 6 labels (drop from 98.1% to 80.6% accuracy). In contrast, our CPC VAE approach is reliable at any model capacity, as it better aligns generative and discriminative goals.

## 2 BACKGROUND: DEEP GENERATIVE MODELS AND SEMI-SUPERVISION

We now describe VAEs as deep generative models and review previous methods for *semi-supervised learning* (SSL) of VAEs, highlighting weaknesses that we later improve upon. We assume all SSL tasks provide two training datasets: an unsupervised (or unlabeled) dataset $\mathcal{D}^U$ of $N$ feature vectors $x$, and a supervised (or labeled) dataset $\mathcal{D}^S$ containing $M$ pairs $(x, y)$ of features $x$ and label $y \in \mathcal{Y}$. Labels are often sparse ($N \gg M$) and can be discrete or continuous.

### 2.1 UNSUPERVISED GENERATIVE MODELING WITH THE VAE

The variational autoencoder (Kingma & Welling, 2014) is an unsupervised model with two components: a generative model and an inference model. The generative model defines for each example a joint distribution $p_\theta(x, z)$ over "features" (observed vector $x \in \mathbb{R}^D$) and "encodings" (hidden vector $z \in \mathbb{R}^C$). The "inference model" of the VAE defines an approximate posterior $q_\phi(z \mid x)$, which is trained to be close to the true posterior ($q_\phi(z \mid x) \approx p_\theta(z \mid x)$) but much easier to evaluate. As in Kingma & Welling (2014), we assume the following conditional independence structure:

$$p_\theta(x, z) = \mathcal{N}(z \mid 0, I_C) \cdot \mathcal{F}(x \mid \mu_\theta(z), \sigma_\theta(z)), \quad q_\phi(z \mid x) = \mathcal{N}(z \mid \mu_\phi(x), \sigma_\phi(x)). \quad (1)$$

The likelihood $\mathcal{F}$ is often multivariate normal, but other distributions may give robustness to outliers. The (deterministic) functions $\mu_\theta$ and $\sigma_\theta$, with trainable parameters $\theta$, define the mean and covariance of the likelihood. Given any observation $x$, the posterior of $z$ is approximated as normal with mean $\mu_\phi$ and (diagonal) covariance $\sigma_\phi$ parameterized by $\phi$. These functions can be represented as *multi-layer perceptrons* (MLPs), *convolutional neural networks* (CNNs), or other (deep) neural networks.

We would ideally learn generative parameters $\theta$ by maximizing the marginal likelihood of features $x$, integrating latent variable $z$. Since this is intractable, we instead maximize a variational lower bound:

$$\max_{\theta, \phi} \quad \sum_{x \in \mathcal{D}} \mathcal{L}^{\text{VAE}}(x; \theta, \phi), \qquad \mathcal{L}^{\text{VAE}}(x; \theta, \phi) = \mathbb{E}_{q_\phi(z|x)} \left[ \log \frac{p_\theta(x, z)}{q_\phi(z|x)} \right] \leq \log p_\theta(x). \quad (2)$$

This expectation can be evaluated via Monte Carlo samples from the inference model $q_\phi(z|x)$. Gradients with respect to $\theta, \phi$ can be similarly estimated by the reparameterization "trick" of representing $q_\phi(z \mid x)$ as a linear transformation of standard normal variables (Kingma & Welling, 2014).

Throughout this paper, we denote variational parameters by $\phi$. Because the factorization of $q$ changes for more complex models, we will write $\phi^{z|x}$ to denote the parameters specific to factor $q(z|x)$.

### 2.2 TWO-STAGE SSL: MAXIMIZE FEATURE LIKELIHOOD THEN TRAIN PREDICTOR

One way to employ the VAE for a semi-supervised task is a *two-stage* "VAE-then-MLP". First, train a VAE to maximize the unsupervised likelihood (2) of all observed features $x$ (both labeled $\mathcal{D}^S$ and unlabeled $\mathcal{D}^U$). Second, we define a label-from-code *predictor* $\hat{y}_w(z)$ that maps each learned code representation $z$ to a predicted label $y \in \mathcal{Y}$. We use an MLP with weights $w$, though any predictor could do. Let $\ell_S(y, \hat{y})$ be a loss function, such as cross-entropy, appropriate for the prediction task. We train the predictor to minimize the loss: $\min_w \sum_{x,y \in \mathcal{D}^S} \mathbb{E}_{q_\phi(z|x)} [\ell_S(y, \hat{y}_w(z))]$. Importantly, this second stage uses only the small labeled dataset and relies on fixed parameters $\phi$ from stage one.

While "VAE-then-MLP" is a simple common baseline (Kingma et al., 2014), it has a key disadvantage: Labels are only used in the second stage, and thus a *misspecified* generative model in stage one will likely produce inferior predictions. Fig. 1 illustrates this weakness.

## 2.3 SEMI-SUPERVISED VAES: MAXIMIZE JOINT LIKELIHOOD OF LABELS AND FEATURES

To overcome the weakness of the two-stage approach, previous work by Kingma et al. (2014) presented a VAE-inspired model called "M2" focused on the *joint* generative modeling of labels $y$ and data $x$. M2 has two components: a generative model $p_\theta(x, y, z)$ and an inference model $q_\phi(y, z \mid x)$. Their generative model is factorized to sample labels (with frequencies $\pi$) first, and then features $x$:

$$p_\theta(x, y, z) = \mathcal{N}(z \mid 0, I_C) \cdot \text{Cat}(y \mid \pi) \cdot \mathcal{F}(x \mid \mu_\theta(y, z), \sigma_\theta(y, z)). \tag{3}$$

The M2 inference model sets $q_\phi(y, z \mid x) = q_{\phi^{y|x}}(y \mid x) q_{\phi^{z|x,y}}(z \mid x, y)$, where $\phi = (\phi^{y|x}, \phi^{z|x,y})$.

To train M2, Kingma et al. (2014) maximize the likelihood of all observations (labels and features):

$$\max_{\theta, \phi^{y|x}, \phi^{z|x,y}} \quad \sum_{x,y \in \mathcal{D}^S} \mathcal{L}^S(x, y; \theta, \phi^{z|x,y}) + \sum_{x \in \mathcal{D}^U} \mathcal{L}^U(x; \theta, \phi^{y|x}, \phi^{z|x,y}). \tag{4}$$

The first, "supervised" term in Eq. (4) is a variational bound for the feature-and-label *joint* likelihood:

$$\mathcal{L}^S(x, y; \theta, \phi^{z|x,y}) = \mathbb{E}_{q_{\phi^{z|x,y}}(z|x,y)} \left[ \log \frac{p_\theta(x,y,z)}{q_{\phi^{z|x,y}}(z|x,y)} \right] \le \log p_\theta(x, y). \tag{5}$$

The second, "unsupervised" term is a variational lower bound for the features-only likelihood $\log p_\theta(x) \ge \mathcal{L}^U$, where $\mathcal{L}^U = \mathbb{E}_{q_\phi(y,z|x)} \left[ \log \frac{p_\theta(x,y,z)}{q_\phi(y,z|x)} \right]$ can be simply expressed in terms of $\mathcal{L}^S$:

$$\mathcal{L}^U(x; \theta, \phi^{y|x}, \phi^{z|x,y}) = \sum_{y \in \mathcal{Y}} q_{\phi^{y|x}}(y \mid x) \left( \mathcal{L}^S(x, y; \theta, \phi^{z|x,y}) - \log q_{\phi^{y|x}}(y \mid x) \right). \tag{6}$$

As with the unsupervised VAE, both terms in the objective can be computed via Monte Carlo sampling from the variational posterior, and gradients can be estimated via the reparameterization trick.

**M2's prediction dilemma and heuristic fix.** After training parameters $\theta, \phi$, we need to predict labels $y$ given test data $x$. M2's structure assumes we make predictions via the inference model's discriminator density $q_{\phi^{y|x}}(y \mid x)$. However, the discriminator's parameter $\phi^{y|x}$ is only informed by the *unlabeled* data when using the objective above (it is not used to compute $\mathcal{L}^S$). We cannot expect accurate predictions from a parameter that does not touch *any* labeled examples in the training set.

To partially overcome this issue, Kingma et al. (2014) and later work use a weighted objective:

$$\max_{\theta, \phi} \sum_{x,y \in \mathcal{D}^S} \left( \alpha \log q_{\phi^{y|x}}(y \mid x) + \lambda \mathcal{L}^S(x, y; \theta, \phi^{z|x,y}) \right) + \sum_{x \in \mathcal{D}^U} \mathcal{L}^U(x; \theta, \phi^{y|x}, \phi^{z|x,y}). \tag{7}$$

This objective biases the inference model's discriminator to do well on the labeled set via an extra loss term (weighted by hyperparameter $\alpha > 0$). We can further include $\lambda > 0$ to balance the supervised and unsupervised terms. Originally, Kingma et al. (2014) fix $\lambda = 1$ and tune $\alpha$ to achieve good performance. Later, Siddharth et al. (2017) tuned $\lambda$ to improve performance. Maaløe et al. (2016) used this same $\alpha \log q(y \mid x)$ term for labeled data to train VAEs with auxiliary variables.

**Disadvantage: What Justification?** While the $\mathcal{L}^S$ and $\mathcal{L}^U$ terms in Eq. (7) have a rigorous justification as maximizing the data likelihood under the assumed generative model, the first term ($\alpha \log q(y \mid x)$) is not justified by the generative or inference model. In particular, suppose the training data were fully labeled: we would ignore the $\mathcal{L}^U$ terms altogether, and the remaining terms would *decouple* the parameters $\theta, \phi^{z|x,y}$ from the discriminator parameters $\phi^{y|x}$. This is deeply unsatisfying: We want a single model guided by *both* generative and discriminative goals, not two separate models. Even in partially-labeled scenarios, including this $\alpha$ term does not adequately balance generative and discriminative goals, as we demonstrate in later examples. An overly flexible yet misspecified generative model may go astray and compromise predictions.

**Disadvantage: Runtime Cost.** Another disadvantage is that the computation of $\mathcal{L}^U$ in Eq. (6) is *expensive*. If labels are discrete, computing this sum exactly is possible but requires a sum over all $L = |\mathcal{Y}|$ possible class labels, computing a Monte Carlo estimate of $\mathcal{L}^S$ for each one. While Monte Carlo approximations can avoid the explicit sum in Eq. (6), they may make gradients too noisy.

**Extensions.** Siddharth et al. (2017) showed how $\mathcal{L}^S$ and $\mathcal{L}^U$ could be extended to any desired conditional independence structure for $q_\phi(y, z \mid x)$, generalizing the label-then-code factorization $q_\phi(y \mid x) q_\phi(z \mid x, y)$ of Kingma et al. (2014). While importance sampling leads to likelihood bounds,

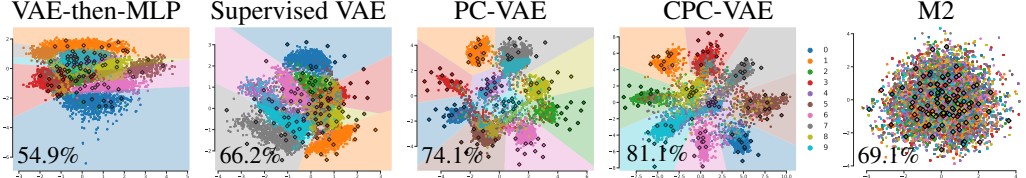

Figure 2: Semi-supervised learning of 2-dim. encodings of MNIST digits, with accuracy in lower corner. All methods use 100 labeled examples and 49,900 unlabeled examples. Each observed image $x$ is mapped to its most likely 2-dim. encoding $z$ and colored by true label $y$. Labeled examples are emphasized. Where applicable, we also show class decision boundaries. *Baselines (from left):* 2-stage unsupervised VAE-then-MLP (Sec. 2.2) and a "supervised" VAE maximizing joint likelihood $\log p(x, y)$ (a special case of our PC method with $\lambda = 1$, Sec. 3.1). *Our methods:* Prediction constrained VAE (PC-VAE with $\lambda = 25$, Sec. 3.1) and consistent prediction constrained VAE (CPC-VAE, Sec. 3.2). *Competitors:* M2 from Kingma et al. (2014), which intentionally decouples label $y$ from "style" $z$, has limited accuracy due to imbalance of discriminative and generative goals.

the overall objective still has two undesirable traits. First, it is expensive, requiring either marginalization of $y$ to compute $\mathcal{L}^U$ in Eq. (6) or marginalization of $z$ to compute $q(y|x) = \int q_\phi(y, z|x)dz$. Second, the approach requires the heuristic inclusion of the discriminator loss $\alpha \log q(y \mid x)$. While recent parallel work by Gordon & Hernández-Lobato (2020) also tries to improve SSL for VAEs, their approach couples discriminative and generative terms only distantly through a joint prior over parameters and still requires expensive sums over labels when computing generative likelihoods.

# 3 PREDICTION-CONSTRAINED LEARNING WITH CONSISTENCY

We now highlight two experiments that demonstrate disadvantages of prior SSL methods, and contrast them with our new approaches. In Fig. 1 we show the predictive accuracy of several SSL methods on the widely-used "half-moon" task, where the goal is to to predict a binary label $y$ given 2-dimensional features $x$. We focus on the top row, which shows results given only 6 labeled examples (3 of each class) but hundreds of unlabeled examples. Notably, while M2 has 98.1% accuracy with a small encoding space ($C = 2$), if the generative model is too flexible ($C = 14$) it learns overly complex structure that does not help label-from-feature predictions, dropping accuracy to only 80.6%. In contrast, our *consistent prediction constrained* (CPC) VAE gets over 98% accuracy with either $C = 2$ or $C = 14$. We have verified it maintains 98% even at $C = 50$, while M2 shows further instability.

Second, in Fig. 2 we show SSL methods for classifying images of MNIST digits (LeCun et al., 2010), given only 10 labeled examples per digit. We seek models with highly accurate label-from-feature predictions, as well as interpretable relationships between the encoding $z$ and these predictions. When forced to use a 2-dimensional latent space, M2 has worse accuracy and (by design) no apparent relationship between encoding $z$ and label $y$. In contrast, our CPC approach offers noticeable advantages over all baselines in both accuracy and interpretability of the encoding space.

## 3.1 PREDICTION CONSTRAINED TRAINING FOR VAES

We develop a framework for *jointly* learning a strong generative model of features $x$, and making label-given-feature predictions $\hat{y}(x)$ of uncompromised quality, by requiring predictions to meet a user-specified quality threshold. Our *prediction constrained* training objective enables end-to-end estimation of all parameters while incorporating the *same* task-specific prediction rules and loss functions that will be used in heldout evaluation ("test") scenarios. Our goals are similar to previous work on end-to-end approximate inference for task-specific losses with simpler probabilistic models (Lacoste-Julien et al., 2011; Stoyanov et al., 2011), but our approach yields simpler algorithms.

**Generative model.** Our generative model does not include labels $y$, only features $x$ and encodings $z$. Their joint distribution $p_\theta(x, z)$ factorizes as the unsupervised VAE of Eq. (1), and we also use the inference model $q_\phi(z \mid x)$ defined in Eq. (1). While M2 included the labels $y$ in its generative model (Kingma et al., 2014), our goals are different: we wish to make label-given-feature predictions, but we are not interested in label marginals or other distributions over $y$ that do not condition on $x$.

**Label-from-feature prediction.** To predict labels $y$ from features $x$, we use a predictor similar to the two-stage method of Sec. 2.2. We first sample an encoding $z \sim q_\phi(z|x)$ from the learned inference

model, and then transform this encoding $z$ to a label via the predictor function $\hat{y}_w(z)$ with parameter $w$. By sharing random variable $z$, the generative model is involved in label-from-feature predictions.

**Constrained PC objective.** Unlike the two-stage model, our approach does not do post-hoc prediction with a previously learned generative model. Instead, we train the predictor simultaneously with the generative model via a new, *prediction-constrained* (PC) objective:

$$\max_{\theta, \phi^{z|x}, w} \sum_{x \in \mathcal{D}^U \cup \mathcal{D}^S} \mathcal{L}^{\text{VAE}}(x; \theta, \phi^{z|x}), \quad \text{subj. to:} \quad \frac{1}{M} \sum_{x,y \in \mathcal{D}^S} \underbrace{\mathbb{E}_{q_\phi(z|x)}[\ell_S(y, \hat{y}_w(z))]}_{\mathcal{P}(x,y;\phi^{z|x},w)} \le \epsilon. \quad (8)$$

The constraint requires that any feasible solution achieve average prediction loss less than $\epsilon$ on the labeled training set. Both the loss function and scalar threshold $\epsilon > 0$ can be set to reflect task-specific needs (e.g., classification must have a certain false positive rate or overall accuracy). The loss function may be any differentiable function, and need not equal the log-likelihood of discrete labels as assumed by previous work specialized to supervision of topic models (Hughes et al., 2018).

**Unconstrained PC objective.** Using the KKT conditions, we define an equivalent unconstrained objective that maximizes the unsupervised likelihood but penalizes inaccurate label predictions:

$$\max_{\theta, \phi^{z|x}, w} \sum_{x \in \mathcal{D}^U \cup \mathcal{D}^S} \mathcal{L}^{\text{VAE}}(x; \theta, \phi^{z|x}) - \lambda \sum_{x,y \in \mathcal{D}^S} \mathcal{P}(x, y; \phi^{z|x}, w). \quad (9)$$

Here $\lambda$ is a positive Lagrange multiplier chosen to ensure that the target prediction constraint is achieved; smaller loss tolerances $\epsilon$ require larger penalty multipliers $\lambda$. This PC objective, and gradients for parameters $\theta, \phi, w$, can be estimated via Monte Carlo samples from $q_\phi(z \mid x)$.

**Justification.** While the PC objective of Eq. (9) may look superficially similar to Eq. (7), we emphasize two key differences. First, our objective couples a generative likelihood and a prediction loss via the shared variational parameters $\phi^{z|x}$. This makes both generative and discriminative performance depend on the *same* learned encoding $z$. (Later we show how to partition $z$ so some entries are discriminative, while others affect generative "style" only.) In contrast, the M2 objective uses a label-given-features conditional to make predictions that does not share *any* of its parameters $\phi^{y|x}$ with the supervised likelihood $\mathcal{L}^S$. Second, our objective is more affordable: no term requires an expensive marginalization over labels. This is key to scaling to big unlabeled datasets, and also enables tractable learning from datasets whose labels are continuous or multi-dimensional.

**Hyperparameters.** The major hyperparameter influencing PC training is the constraint multiplier $\lambda \ge 0$. Setting $\lambda = 0$ leads to unsupervised maximum likelihood training (or MAP training, given priors on $\theta$) of a classic VAE. Setting $\lambda = 1$ and choosing a probabilistic loss $-\log p(y \mid z)$ produces a "supervised VAE" that maximizes the joint likelihood $p_\theta(x, y)$. But as illustrated in Fig. 2, because features $x$ have much higher dimension than labels $y$, the resulting model may have weak predictive performance. Satisfying the strong prediction constraint of Eq. (8) typically requires $\lambda \gg 1$, and in practice we use validation data to select the best of several candidate $\lambda$ values. If a task motivates a concrete tolerance $\epsilon$, we can test an increasing sequence of $\lambda$ values until the constraint is satisfied.

We emphasize that although Eq. (9) is easier to optimize, we prefer to think of the constrained problem in Eq. (8) as the "primary" objective, because our applied goals are to satisfy discriminative quality first; a generative model that predicts poorly is not plausible. Furthermore, the constrained objective is far more natural for semi-supervised learning. The choice of $\epsilon$ need not be concerned by the relative sizes of the labeled and unlabeled datasets. In contrast, if either $|\mathcal{D}^U|$ or $|\mathcal{D}^S|$ changes, the value of $\lambda$ may need to change dramatically to reach the same prediction quality.

## 3.2 ENFORCING CONSISTENT PREDICTIONS FROM GENERATIVE MODEL RECONSTRUCTIONS

While the PC objective is effective given sufficient labeled data, it may generalize poorly when labels are very sparse (see Fig. 1). This fundamental problem arises because in the PC objective of Eq. (8), the parameters $w$ of the predictor $\hat{y}_w(z)$ are only directly informed by the labeled training data.

Revisiting the generative model, let $x \sim p_\theta(\cdot \mid z')$ and $\bar{x} \sim p_\theta(\cdot \mid z')$ be two observations sampled from the *same* latent code $z'$. Even if the true label $y$ of $x$ is uncertain, we know that for this model to be useful for predictive tasks, $\bar{x}$ must have the *same* label as $x$. We formalize this relationship via a *consistency constraint* requiring label predictions for common-code data pairs $(x, \bar{x})$ to approximately match (see Fig. 3). As we show, this regularization may dramatically boost performance.

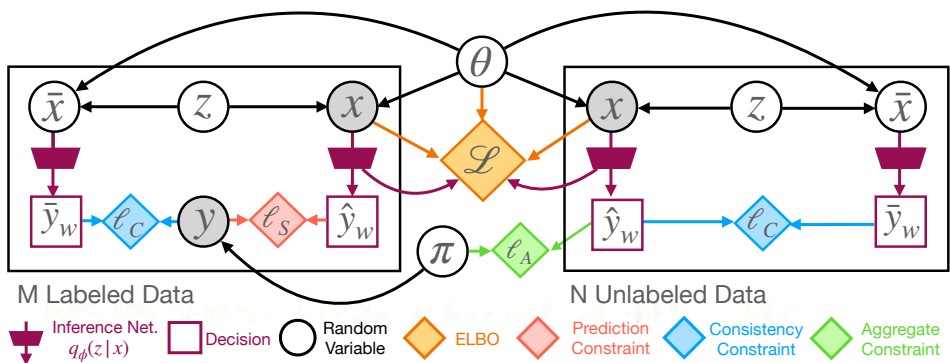

Figure 3: Formalization of our consistency-constrained VAE as a decision network (Cowell et al., 2006). Circular nodes are random variables, including the latent VAE code $z$ that generates observed features $x$. Shaded nodes are observed, including the class labels $y$ for some data (left). Square decision nodes indicate predictions $\hat{y}_w$ of class labels that depend on the amortized inference network $q_\phi(z \mid x)$. Diamonds indicate losses (negative utilities) that influence the variational prediction of labels and latent variables. *Generative likelihood:* Like standard VAEs, our generalizations choose generative model parameters $\theta$ and variational posteriors $q_\phi$ to maximize the ELBO $\mathcal{L}$ (*orange*). *Prediction accuracy:* Unlike previous semi-supervised VAEs, we do not directly model the probabilistic dependence of labels $y$ on $z$ and/or $x$. We instead treat label prediction as a decision problem, with application-motivated loss $\ell_S$ (*red*), that constrains the approximate VAE posterior $q_\phi(z|x)$ (and therefore the encodings and generative model). *Prediction consistency:* For unlabeled data (right), we cannot directly evaluate the quality of predictions. However, we do know that if two observations $x$ and $\bar{x}$ are generated from the same latent code $z$, they should have identical labels; otherwise, the model cannot have high accuracy. The loss $\ell_C$ (*blue*) enforces the consistency of such predictions. *Aggregate consistency:* By the law of large numbers, we also know that aggregate label frequencies for unlabeled data should be close to the frequencies $\pi$ observed in labeled data. The loss $\ell_A$ (*green*) enforces this constraint, and penalizes degenerate predictors that satisfy $\ell_C$ by predicting the same label $\hat{y}_w$ for most or all of the unlabeled data.

Given features $x$, our method predicts labels by sampling $z \sim q_\phi(z \mid x)$ from the approximate posterior, and then applying our predictor $\hat{y}_w(z)$. Alternatively, given $x$ we can first *simulate* alternative features $\bar{x}$ with matching code $z$ by sampling from the inference and generative models, and then predict the label associated with $\bar{x}$. We constrain the label predictions $y$ for $x$, and $\bar{y}$ for $\bar{x}$, to be similar via a consistency penalty function $\ell_C(y, \bar{y})$. For the classification tasks considered below, we use a cross-entropy consistency penalty $\ell_C$. Given this penalty, we constrain the maximum values of the following consistency costs on *unlabeled* and *labeled* examples, respectively:

$$\mathcal{C}^U(x; \theta, \phi, w) \triangleq \mathbb{E}_{q_\phi(z|x)} \left[ \mathbb{E}_{p_\theta(\bar{x}|z)} \left[ \mathbb{E}_{q_\phi(\bar{z}|\bar{x})} \left[ \ell_C(\hat{y}_w(z), \hat{y}_w(\bar{z})) \right] \right] \right], \tag{10}$$

$$\mathcal{C}^S(x, y; \theta, \phi, w) \triangleq \mathbb{E}_{q_\phi(z|x)} \left[ \mathbb{E}_{p_\theta(\bar{x}|z)} \left[ \mathbb{E}_{q_\phi(\bar{z}|\bar{x})} \left[ \ell_C(y, \hat{y}_w(\bar{z})) \right] \right] \right]. \tag{11}$$

**Consistent PC: Unconstrained objective.** To train parameters, we apply our consistency costs to unlabeled and labeled feature vectors, respectively. The overall objective becomes:

$$\max_{\theta, \phi, w} \sum_{x \in \mathcal{D}^U \cup \mathcal{D}^S} \mathcal{L}^{\text{VAE}}(x; \theta, \phi) - \sum_{x \in \mathcal{D}^U} \gamma \mathcal{C}^U(x; \theta, \phi, w) + \sum_{x, y \in \mathcal{D}^S} -\lambda \mathcal{P}(x, y; \phi, w) - \gamma \mathcal{C}^S(x, y; \theta, \phi, w),$$

where $\mathcal{L}$ is the unsupervised likelihood, $\mathcal{P}$ is the predictor loss, and $\mathcal{C}$ are the consistency constraints. Here, $\gamma > 0$ is a scalar Lagrange multiplier for the consistency terms, with similar interpretation as $\lambda$.

**Aggregate Label Consistency.** For SSL applications, we find it is also useful to regularize our model with an *aggregate label consistency* constraint, which forces the distribution of label predictions for unlabeled data to be aligned with a known target distribution $\pi$. This discourages predictions on ambiguous unlabeled examples from collapsing to a single value. We define the aggregate consistency loss as: $\ell_A\big(\pi, \mathbb{E}_{x \sim \mathcal{D}^U, z \sim q(z|x)}[\hat{y}_w(z)]\big)$, and again use a cross-entropy penalty. If the target distribution of labels $\pi$ is unknown, we set it to the empirical distribution of the labeled data.

**Related work on consistency.** Recently popular SSL image classifiers focused on discriminative goals will train the weights of a CNN to minimize a modified objective that penalizes both label accuracy and a notions of consistency or smoothness on unlabeled data. Examples include consistency under adversarial perturbations (Miyato et al., 2019), label-invariant transformations (Laine & Aila, 2017), and when interpolating between training features (Berthelot et al., 2019). This regularization can deliver competitive discriminative performance, but does not meet our goal of generative modeling.

Recently, Unsupervised Data Augmentation (UDA, Xie et al. (2020)), achieved state-of-the-art vision and text SSL classification by enforcing label consistency on augmented samples of unlabeled features. UDA relies on the availability of well-engineered augmentation routines for specific domains (e.g. image processing library transforms for vision or back-translation for text). In contrast, we learn a *generative* model that produces feature vectors for which predictions need to be consistent. Our approach is more applicable to new domains where advanced augmentation routines are not available.

In broader machine learning, "cycle-consistency" has improved generative adversarial methods for images (Zhu et al., 2017; Zhou et al., 2016) or biomedical data (McDermott et al., 2018). Others have developed cycle-consistent objectives for VAEs (Jha et al., 2018) which focus on consistency in code vectors $z$. In contrast, our work focuses on semi-supervised learning and enforces cycle consistency in labels $y$. Recently, Miller et al. (2019) developed discriminative regularization for VAEs. Their objective is not designed for SSL and uses a direct feature-to-label prediction model that must be consistent with reconstructed predictions. Our approach uses code-to-label prediction and SSL.

### 3.3 IMPROVED GENERATIVE MODELS: ROBUST LIKELIHOODS AND SPATIAL TRANSFORMERS

As our approach is applicable to any generative model, we can incorporate prior knowledge of the data domain to improve both generative and discriminative performance. We consider two examples: likelihoods that model noisy pixels, and explicit affine transformations to model image deformations.

**Robust Likelihoods.** Instead of a normal (or other common) likelihood we use a "Noise-Normal" likelihood to model images more robustly. We assume that pixel intensities $x$ have values in the interval $[-1, 1]$ and rescale our datasets to match. Our Noise-Normal likelihood is defined as a 2-component mixture of a truncated Normal and a uniform "noise" distribution, with pixel-specific mixture weights. Define the standard normal PDF as $\phi(\cdot)$ and standard normal CDF as $\Phi(\cdot)$. We write the *probability density function* of the Noise-Normal distribution with parameters $(\rho, \mu, \sigma)$ as:

$$\mathcal{F}(x \mid \rho, \mu, \sigma) = \rho \left( \frac{\phi\left(\frac{x-\mu}{\sigma}\right)}{\Phi\left(\frac{1-\mu}{\sigma}\right) - \Phi\left(\frac{-1-\mu}{\sigma}\right)} \right) + (1 - \rho)\left(\frac{1}{2}\right). \tag{12}$$

Following Eq. (1), our (unsupervised) VAE now uses the revised generative and inference models:

$$p_\theta(x, z) = \mathcal{N}(z \mid 0, I_C) \cdot \mathcal{F}(x \mid \rho_\theta(z), \mu_\theta(z), \sigma_\theta(z)), \quad q_\phi(z \mid x) = \mathcal{N}(z \mid \mu_\phi(x), \sigma_\phi(x)). \tag{13}$$

This approach allows our model to avoid sensitivity to outliers and noise in the observed images, and boosts the performance of our CPC method for SSL.

**Spatial Transformer VAE.** Our spatial transformer VAE retains the structure of a standard VAE, but reinterprets the latent code $z$ as two components. We denote the first 6 latent dimensions as $z_t$, and associate these with 6 affine transformation parameters capturing image translation, rotation, scaling, and shear. The generative model maps each value into a fixed range and creates an affine transformation matrix, $M_t$, by applying the transformations in a fixed order. The remainder of the latent code, $z_*$, is used to generate parameters for independent, per-pixel likelihoods. Assuming normal likelihoods, the output parameters for the pixel with coordinates $(i, j)$ are $\mu_\theta(z_*)_{ij}, \sigma_\theta(z_*)_{ij}$.

We re-orient our per-pixel likelihoods according to the affine transform $M_t$. The parameters of the likelihood for pixel $(i, j)$ will use the decoder outputs at coordinate $(i', j')$, where $M_t$ defines the affine mapping from $(i', j')$ to $(i, j)$. We apply this transformation in a (sub) differentiable way via a *spatial transformer layer* (Jaderberg et al., 2015) that takes as input $M_t$ and the appropriate parameter maps $\mu_\theta(z_*), \sigma_\theta(z_*)$, and outputs a final set of parameters for the individual pixel likelihoods. As $(i', j')$ may not correspond to integer coordinates, we use bilinear interpolation over an appropriate representation of the likelihood parameters, and appropriately pad the size of the decoder output.

We further account for this special structure in our prediction and consistency constraints. For many applications, we have prior knowledge that small affine transforms should not affect the class of an image, and thus we can define consistency constraints that condition on $z_*$ but not $z_t$.

## 4 EXPERIMENTS

We assess our consistent prediction-constrained (CPC) VAE on two key goals: accurate prediction of labels $y$ given features $x$ (especially when labels are rare) and useful generative modeling of $x$. We compare to ablations of our own method (without consistency, without spatial transformations) and to external baselines. We report each method's mean and standard deviation in classification accuracy

| Source | Method | MNIST (100) | SVHN (1000) | NORB (1000) |
|---|---|---|---|---|
| Tables 1-2 of Kingma et al. (2014) | M1 + M2 | 96.67% ($\pm 0.14$) | 63.98% ($\pm 0.10$) | - |
| Table 2 of Maaløe et al. (2016) | ADGM | **99.04**% ($\pm 0.02$) | 77.14% | 89.94% ($\pm 0.05$) |
| Table 2 of Maaløe et al. (2016) | SDGM | 98.68% ($\pm 0.07$) | 83.39% ($\pm 0.24$) | 90.60% ($\pm 0.04$) |
| Gordon & Hernández-Lobato (2020) | Blended M2 | 93.05% ($\pm 0.73$) | - | - |
| Tables 3-4 of Miyato et al. (2019) | VAT | 98.64% ($\pm 0.03$) | **94.23**% ($\pm 0.32$) | - |
| ours, using labeled-set only | WRN | 73.91% ($\pm 1.45$) | 87.7% ($\pm 1.02$) | 86.7% ($\pm 1.32$) |
| ours | **CPC VAE** | 98.29% ($\pm 0.50$) | **94.22**% ($\pm 0.62$) | **92.00**% ($\pm 1.21$) |

Table 1: Test accuracy of SSL methods. Our results show mean (std. dev.) across 10 samples of the labeled set. The labeled-set-only discriminative neural net (WRN) has roughly the same size as our CPC VAE.

| Method | MNIST (100) | | Method | MNIST (100) |
|---|---|---|---|---|
| **CPC** (2 layer) | **96.68**% ($\pm 0.54$) | | **M1 + M2** (Kingma et al., 2014) | **96.67**% ($\pm 0.14$) |
| CPC (2 layer, w/o aggregate loss) | 94.27% ($\pm 3.78$) | | M2 (1 layer, $\alpha = 0.1$) (Kingma et al., 2014) | 88.03% ($\pm 1.71$) |
| CPC (2 layer, w/o transforms) | 91.93% ($\pm 1.65$) | | M2 (2 layer, $\alpha = 0.1$) | 83.32% ($\pm 5.22$) |
| CPC (4 layer, w/o transforms) | 93.78% ($\pm 2.25$) | | M2 (4 layer, $\alpha = 0.1$) | 47.05% ($\pm 8.13$) |
| PC (2 layer) | 80.49% ($\pm 3.31$) | | M2 (4 layer, tuned to $\alpha = 10$) | 68.15% ($\pm 3.43$) |
| VAE + MLP | 72.90% ($\pm 1.98$) | | M2 (1 layer, $\alpha = 0.1$, Noise-Normal) | 73.93% ($\pm 8.12$) |

Table 2: Ablation study for SSL on MNIST using a dense MLP matching M2's network architecture. Trials used 100 labels and encoding size $C = 50$. Unless cited, all results come from our implementation, where encoder and decoder have 1000 units per hidden layer. All our introduced techniques improve accuracy. M2's accuracy deteriorates when its networks are overparameterized, even after tuning $\alpha$ on validation set instead of using default from Kingma et al. (2014), while our method remains stable.

across 10 labeled subsets. We trained using ADAM (Kingma & Ba, 2014), with each minibatch containing 50% labeled and 50% unlabeled data. Hyperparameter search used Optuna (Akiba et al., 2019) to maximize accuracy on validation data. Supervised baselines used either MLP or wide residual nets (WRN, Zagoruyko & Komodakis (2016)). Reproducible details are in appendices.

**SSL classification on MNIST with thorough internal comparisons.** In Table 2 we compare several variations of our CPC methods and the M2 model on an SSL version of MNIST (LeCun et al., 2010, 10 classes, $|\mathcal{D}^S| = 100, |\mathcal{D}^U| = 49900$, 10000 validation, 10000 test).

**SSL classification on SVHN and NORB.** Table 1 compares our methods on two standard SSL tasks: Street-View Housing Numbers (SVHN) (Netzer et al., 2011, 10 classes, $|\mathcal{D}^S| = 1000, |\mathcal{D}^U| = 62257$, 10000 validation, 26032 test) and the NYU Object Recognition Benchmark (NORB) (LeCun et al., 2004, 5 classes, $|\mathcal{D}^S| = 1000, |\mathcal{D}^U| = 21300$, 2000 validation, 24300 test).

**SSL classification on CelebA.** We ran additional experiments on a variant of the CelebA dataset (Liu et al., 2015). For these trials we created a classification problem with 4 classes based on the combination of gender (woman/man) and facial expression (neutral/smiling). (4 classes, $|\mathcal{D}^S| = 1000$, $|\mathcal{D}^U| = 159770$, 2000 validation, 19962 test). We report our results in figure 6.

Across all evaluations, we can conclude:

**Both consistency and prediction constraints are needed for high accuracy.** In Table 2, PC alone gets 80% accuracy on 100-label MNIST, while adding consistency yields 97% for CPC. The benefits of CPC over PC in both accuracy and latent interpretability are visible in Figs. 1-2. Our *aggregate label consistency* improves robustness, reducing the *variance* in CPC accuracy (from $3.78^2$ to $0.54^2$).

**CPC training delivers strong improvements in SSL prediction quality over baselines.** In Table 1, our CPC achieves 94.22% and 92.0% on the challenging 1000-label SVHN and NORB benchmarks, which surpasses by >1.4% all reported baselines while being *reliable* across runs. The M1+M2 baseline (Kingma et al., 2014) is not a coherent generative model, but rather a discriminative ensemble of multiple models. It performs well on MNIST, but very poorly on the more challenging SVHN.

**CPC delivers better generative performance; it is not all about prediction.** We improve on unsupervised VAEs by explicitly learning latent representations informed by class labels (Fig. 2). In Fig. 4, Fig. 5, and Fig. 6, we show visually-plausible class-conditional samples from our best CPC models. Additional visuals from learned VAE and CPC-VAE models are in the supplement.

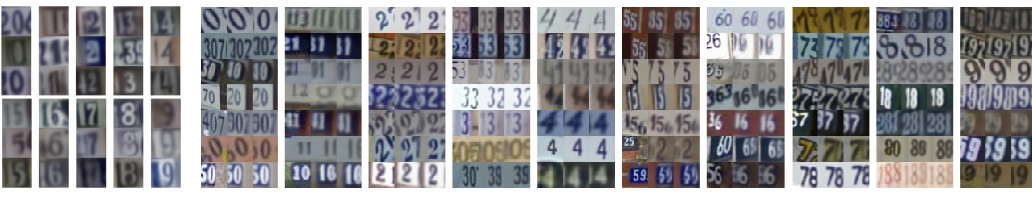

Figure 4: Sampled reconstructions used to compute the consistency loss during training. *Top:* Original image. *Middle:* Sampled reconstructions using a "Noise-Normal" likelihood. *Bottom:* Sampled reconstructions with spatial affine transformations sampled from the prior.

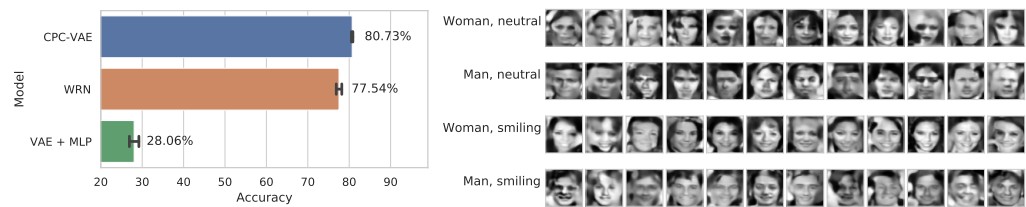

(a) Prior samples                               (b) Reconstructions

Figure 5: Visualizations of generative model performance on SVHN for a prediction and consistency constrained VAE incorporating latent affine transformations, trained on the SVHN dataset with all labels. (5a) Samples from the learned generative model conditioned on class (more are shown in supplement B). Samples are chosen via rejection sampling in the latent space with a threshold of 95% confidence in the target class. (5b) Reconstructions of images in the held-out test set. Each triplet shows the original image (left), the reconstruction (middle), and an aligned reconstruction (right) obtained by fixing the learned affine transform variables to the global mean.

Figure 6: CelebA dataset results. *Left:* Test accuracy on the 4-class CelebA SSL task (1000 labeled training images, 159,770 unlabeled; error bars show std. dev. of 10 labeled set samples). CPC-VAE improves on the labeled-set only WRN, and especially beats the unsupervised VAE which poorly separates the classes in the latent space. *Right:* Class-conditional samples of the 4 possible classes from our semi-supervised CPC VAE.

**With improved generative models, CPC can improve predictions.** Fig. 5 shows that including spatial transformations allows learning a canonical orientation and scale for each digit. This generative improvement boosts classifier accuracy (e.g., MNIST improves from 91.9% to to 96.7% in Table 2).

## 5  CONCLUSION

We have developed a new optimization framework for semi-supervised VAEs that can balance discriminative and generative goals. Across image classification tasks, our CPC-VAE method delivers superior accuracy and label-informed generative models with visually-plausible samples. Unlike previous efforts to enforce constraints on latent variable models, such as expectation constraints (Mann & McCallum, 2010), posterior regularization (Zhu et al., 2014; 2012), posterior constraints (Ganchev et al., 2010), or prediction constraints for topic models (Hughes et al., 2018), our approach is the only one that coherently and simultaneously treats *uncertainty* in latent variables $z$, applies to flexible "deep" non-conjugate models, and offers scalable training and test evaluation via amortized inference.

A further contribution is demonstrating the necessity of *consistency* for improving discrimination. Our CPC approach is an antidote to model misspecification: the constraints on prediction quality and consistency prevent the model from learning a generative model that is unaligned with the classification task or that overfits with more flexible generative models (as M2 is vulnerable to do). As we show with spatial transformers, our work lets improvements in generative model quality directly improve semi-supervised label prediction, helping realize the promise of deep generative models.

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

# A    CODE AVAILABILITY

During the review process, anonymized code for our methods is available. After review, we will publish all code under a permissive open source license.

Download a ZIP file here: `https://www.dropbox.com/s/b43xcsnhs5kevue/PC-VAE-REVIEW-RELEASE.zip?dl=1`

Browse in your web browser here: `https://anonymous.4open.science/repository/0fa4ef53-5e92-40df-ad0a-bcdab28f6df0/pcvae/`

# B    DETAILS AND VISUALIZATIONS OF GENERATIVE MODELS

## B.1    NOISE-NORMAL LIKELIHOOD

As discussed in Sec. 4.3, we use a "Noise-Normal" distribution as the pixel likelihood for many of our experiments. We define this distribution to be a parameterized two-component mixture of a *truncated-normal* distribution and a *uniform* distribution. We will use $\rho$ to denote the mixture probability of the Normal component, and $\mu$ and $\sigma$ to denote the mean and standard deviation of the truncated-normal, respectively. The generative model (or decoder) predicts a distinct outlier probability $(1 - \rho)$ for each pixel. We assume that pixel intensities are defined on the domain $[-1, 1]$ and rescale our datasets to match. We can write the *probability density function* of the Noise-Normal distribution via the standard normal PDF $\phi(\cdot)$, and standard normal CDF $\Phi(\cdot)$, as follows:

$$f(x \mid \rho, \mu, \sigma) = \rho \left( \frac{\phi \left( \frac{x-\mu}{\sigma} \right)}{\Phi \left( \frac{1-\mu}{\sigma} \right) - \Phi \left( \frac{-1-\mu}{\sigma} \right)} \right) + (1 - \rho) \left( \frac{1}{2} \right). \tag{14}$$

We can similarly express the *cumulative distribution function* of the Noise-Normal distribution as:

$$F(x \mid \rho, \mu, \sigma) = \rho \left( \frac{\Phi \left( \frac{x-\mu}{\sigma} \right) - \Phi \left( \frac{-1-\mu}{\sigma} \right)}{\Phi \left( \frac{1-\mu}{\sigma} \right) - \Phi \left( \frac{-1-\mu}{\sigma} \right)} \right) + (1 - \rho) \left( \frac{x+1}{2} \right). \tag{15}$$

In order to propagate gradients through the sampling process of the noise-normal distribution, we use the *implicit reparameterization gradients* approach of Figurnov et al. (2018). Given a sample $x$ drawn from this distribution, we compute the gradient with respect to the parameters $\rho$, $\mu$, and $\sigma$ as:

$$\nabla_{\rho,\mu,\sigma} x = \frac{-\nabla_{\rho,\mu,\sigma} F(x \mid \rho, \mu, \sigma)}{f(x \mid \rho, \mu, \sigma)}. \tag{16}$$

When fitting the parameters of this distribution using gradient descent, we enforce the constraints that $\rho \in [0, 1]$, $\mu \in [-1, 1]$, and $\sigma > 0$. To do this, we optimize unconstrained parameters $\rho_*, \mu_*, \sigma_*$, and then define $\rho = \text{sigmoid}(\rho_*)$, $\mu = \tanh(\mu_*)$, and $\sigma = \text{softplus}(\sigma_*)$.

## B.2    SPATIAL TRANSFORMER VAE

We now describe how to sample affine transformations $M_t$ for use in our generative model of images. As described in Sec. 4.3, the latent transformation code $z_t$ has 6 real-valued dimensions, each corresponding to one of the following 6 affine transformation parameters:

- $z_t^{(1)} \rightarrow$ *horizontal translation*,
- $z_t^{(2)} \rightarrow$ *vertical translation*,
- $z_t^{(3)} \rightarrow$ *rotation*,
- $z_t^{(4)} \rightarrow$ *shear*,
- $z_t^{(5)} \rightarrow$ *horizontal scale*,
- $z_t^{(6)} \rightarrow$ *vertical scale*.

To constrain our transformations to a fixed range of plausible values, we construct $M_t$ using parameters $\bar{z}_t^{(i)} = \tanh(z_t^{(i)})$ that are first mapped to the interval $[-1, +1]$, and then linearly rescaled to an appropriate range via hyperparameters $\alpha^{(1)}, \ldots, \alpha^{(6)}$. Figure 7 illustrates that the induced prior for

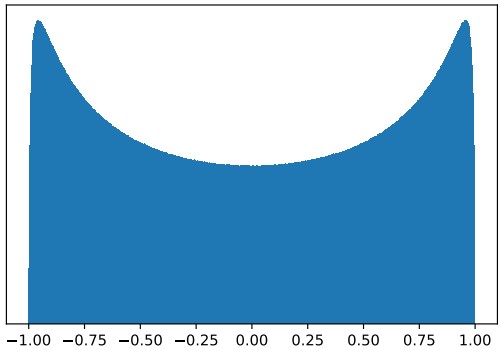

Figure 7: Prior distribution for latent parameters $\bar{z}_t^{(i)} = \tanh(z_t^{(i)})$ used to represent affine transformations.

$\bar{z}_t^{(i)}$ is heaviest for extreme values, encouraging aggressive augmentation when sampling from the prior. The mapping function could be changed to modify this distribution for other applications.

Given these latent transformation parameters, we define an affine transformation matrix $M_t$ as follows:

$$M_t = \begin{bmatrix} 1 & 0 & \alpha^{(1)}\bar{z}_t^{(1)} \\ 0 & 1 & \alpha^{(2)}\bar{z}_t^{(2)} \\ 0 & 0 & 1 \end{bmatrix} \cdot \begin{bmatrix} \cos(\alpha^{(3)}\bar{z}_t^{(3)}) & -\sin(\alpha^{(3)}\bar{z}_t^{(3)}\alpha^{(4)}\bar{z}_t^{(4)}) & 0 \\ \sin(\alpha^{(3)}\bar{z}_t^{(3)}) & \cos(\alpha^{(3)}\bar{z}_t^{(3)}\alpha^{(4)}\bar{z}_t^{(4)}) & 0 \\ 0 & 0 & 1 \end{bmatrix} \cdot \begin{bmatrix} (\alpha^{(5)})^{\bar{z}_t^{(5)}} & 0 & 0 \\ 0 & (\alpha^{(6)})^{\bar{z}_t^{(6)}} & 0 \\ 0 & 0 & 1 \end{bmatrix} \quad (17)$$

To determine the parameters of the likelihood function for the pixel at coordinate $(i, j)$, we use the generative model (or decoder) output at the pixel $(i', j')$ for which

$$\begin{bmatrix} i \\ j \\ 1 \end{bmatrix} = M_t \begin{bmatrix} i' \\ j' \\ 1 \end{bmatrix}. \quad (18)$$

This corresponds to applying horizontal and vertical scaling, followed by rotation and shear, followed by translation. We use the *spatial transformer layer* proposed by Jaderberg et al. (2015) with bilinear interpolation to apply this transformation with non-integer pixel coordinates. For the Noise-Normal distribution we independently interpolate the $\rho$, $\mu$, and $\sigma^2$ parameters.

### B.3 CLASS-CONDITIONAL SAMPLING

A standard VAE generates data by sampling $z \sim \mathcal{N}(0, I)$, and then sampling $x \sim \mathcal{N}(\mu_\theta(z), \sigma_\theta(z))$, or an alternative like the Noise-Normal likelihood. For the PC-VAE or CPC-VAE, we can further sample images conditioned on a particular class label. As labels are not explicitly part of the generative model, we accomplish this by sampling images that would be confidently predicted as the target class. We use a rejection sampler, repeatedly sampling $z \sim \mathcal{N}(0, I)$ until a sample meets the criteria: $p_w(y \mid z) > 1 - \epsilon$, for some target threshold $\epsilon$. We typically use $\epsilon = 0.05$ in our experiments.

**MNIST digit samples for models with a 2-D latent space.** Fig. 2 in the main text shows 2-dimensional latent space encodings of the MNIST dataset using several different models. We provide a complementary visualization of generative models in Fig. 8, where we compare class-conditional samples for three of these models. The unsupervised VAE's encodings of some classes (e.g., 2's and 4's and 8's and 9's) are not separated, and samples thus frequently appear to be the wrong class. Model M2 (Kingma et al., 2014) explicitly encodes the class label as a latent variable, but nevertheless many sampled images do not visually match the conditioned class. In contrast, for our CPC-VAE model almost all samples are easily recognized as the target class.

## C SENSITIVITY TO CONSTRAINT MULTIPLIERS

We compare the test accuracy for our consistency-constrained model for MNIST over a range of values for both $\lambda$ (the prediction constraint multiplier) and $\gamma$ (the consistency constraint multiplier) in

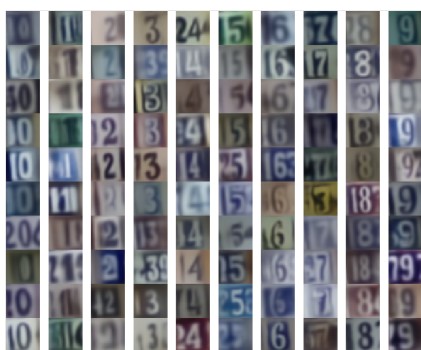

(a) Unsupervised VAE          (b) CPC-VAE          (c) M2

Figure 8: Class-conditional samples of the 10 possible digit classes in the MNIST dataset. Each column shows multiple samples from one specific digit class. From left to right, each panel shows samples from a standard unsupervised VAE, our CPC-VAE, and model M2 (Kingma et al., 2014). All models use a 2-dimensional latent code, and are trained on the MNIST dataset with 100 labeled examples (10 per class).

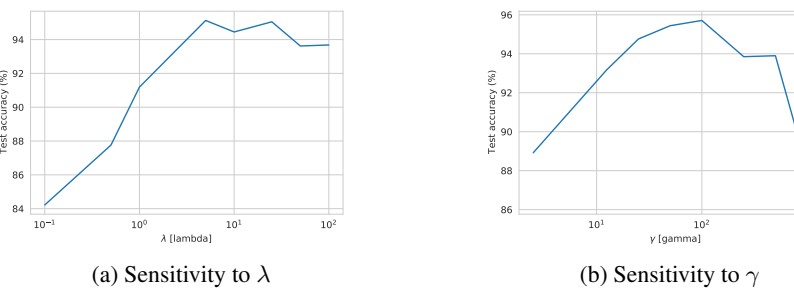

Figure 9: Class-conditional samples of the 10 possible digit classes in the SVHN dataset, extending figure 5a. The generative model was trained on the fully labeled SVHN dataset with prediction and consistency constraints. Samples were chosen via rejection sampling in the latent space with a threshold of 95% confidence in the target class.

(a) Sensitivity to $\lambda$          (b) Sensitivity to $\gamma$

Figure 10: Sensitivity of test accuracy to the constraint (lagrange multiplier) hyperparameters $\lambda$ and $\gamma$

figure 10. All runs used our best consistency-constrained model for MNIST using dense networks. We kept all hyperparameters identical to the previous results (see section D), changing only the value of interest for each run.

We see that the resulting test accuracy smoothly varies across several orders of magnitude, with the optimal result being at or near the values we chose for our experiments.

## D EXPERIMENTAL PROTOCOL

Here we provide details about models and experiments which did not fit into the primary paper.

### D.1 HYPERPARAMETER OPTIMIZATION

The hyperparameter search for all models, including the CPC-VAE and various baselines, used Optuna (Akiba et al., 2019) to achieve the best accuracy on a labeled validation set. For our 2-layer and 4-layer M2 experiments, we used our own implementation (available in our code release) and followed the hyperparameters used by the original authors. For the 4-layer variant, we tested 10 different settings of $\alpha$, ranging from 0.05 to 50, reporting both the the result using the original suggested value for $\alpha$ (0.1) and the best value for $\alpha$ we found for the setting (10). For M2, we also dynamically reduced the learning rate when the validation loss plateaued.

### D.2 NETWORK ARCHITECTURES

For our PC-VAE and CPC-VAE models of the MNIST data, we use fully-connected encoder and decoder networks with two hidden layers, 1000 hidden units per layer, and softplus activation functions. Like the M2 model (Kingma et al., 2014), we use a 50-dimensional latent space. The original M2 experiments used networks with a single hidden layer of 500 units. We compare this to replications with networks matching ours, as well as 4-layer networks.

For the SVHN and NORB datsets, we adapt the wide-residual network architecture (WRN-28-2) (Zagoruyko & Komodakis, 2016) that was proposed as a standard for semi-supervised deep learning research in Oliver et al. (2018). In particular, we use this architecture for our encoder with two notable changes: We replace the final global average pooling layer with 3 dense layers with 1000 hidden units, and add a final dense layer that outputs means and variances for the latent space. We find that the dense layers provide the capacity needed for accomplishing both generative and discriminative tasks with a single network. For the decoder network we use a "mirrored" version of this architecture, reversing the order of layer sizes used, replacing convolutions with transposed convolutions, and removing pooling layers. We maintain the residual structure of the network. Our best classification results with this architecture were achieved with a latent space dimension of 200.

### D.3 BETA-VAE REGULARIZATION

As an additional form of regularization for our model, we allow our hyperparameter optimization to adjust a weight on the KL-divergence term in the variational lower bound, which we call $\boldsymbol{\beta}$ as in previous work (Higgins et al., 2017):

$$\mathcal{L}_{\beta}^{\text{VAE}}(x; \theta, \phi) = \mathbb{E}_{q_{\phi}(z|x)} \left[\log p_{\theta}(x \mid z)\right] + \boldsymbol{\beta} \cdot \mathbb{E}_{q_{\phi}(z|x)} \left[\log \frac{p(z)}{q_{\phi}(z \mid x)}\right] \tag{19}$$

This allows us to encourage $q_{\phi}(z \mid x)$ to more closely conform to the prior, which may be necessary to balance the scale of the objective, depending on the likelihoods used and the dimensionality of the dataset.

### D.4 PREDICTION MODEL REGULARIZATION

We add two standard regularization terms to the prediction model used in our constraint, $\hat{y}_w(y \mid z)$. The first is an $\ell_2$ regularizer on the regression weights, $||w||_2^2$, to help reduce overfitting. The second is an entropy penalty. As $\hat{y}_w(z)$ defines a categorical distribution over labels, we compute this as: $-\mathbb{E}_{\hat{y}_w(y|z)}[\log \hat{y}_w(y \mid z)]$, which has been shown to be helpful for semi-supervised learning in Grandvalet & Bengio (2004) and was used as part of the standardized training framework of Oliver et al. (2018). We allowed our hyperparameter optimization approach to select appropriate weights for both terms.

### D.5 IMAGE PRE-PROCESSING

For all of our image datasets, we rescale the inputs to the range [-1, 1]. For our NORB classification results, we downsample each image to 48x48 pixels. For our SVHN classification results, we convert

images to greyscale to reduce the representational load on our generative model. Before the grayscale conversion, we apply contrast normalization to better disambiguate the colors within each image.

For the SVHN and NORB results, we follow the recommendation of a recent survey of semi-supervised learning methods (Oliver et al., 2018) and apply a single data augmentation technique: random translations by up to 2-pixels in each direction. For generative results, we retained the original color images and trained with full labels.

### D.6 Likelihoods

For all of our image datasets, we use the Noise-Normal likelihood for our CPC methods. For all experiments on toy data (e.g. half-moon), we used a normal likelihood.

For our implementation of M2 for extensive experiments on MNIST we retained the Bernoulli likelihood used by the original authors (Kingma et al., 2014). That is, we rescaling each pixel's numerical intensity value to the unit interval [0,1], and then sampled binary values from a Bernoulli with probability equal to the intensity.

### D.7 Summary of hyperparameter settings for final results

Table 3 below provides all hyperparameter settings used in our experiments.

| Hyperparameter | MNIST (100) | SVHN (1000) | NORB (1000) |
|---|---|---|---|
| Encoder/decoder | 2 FC layers | WRN-28-2 + 3 FC | WRN-28-2 + 3 FC |
| Fully connected layer size | 1000 units | 1000 units | 1000 units |
| Network activations | Softplus | Leaky ReLU | Leaky ReLU |
| Latent dimension | 50 | 200 | 200 |
| Pixel likelihood | Noise-Normal | Noise-Normal | Noise-Normal |
| Prediction multiplier $\lambda$ | 25 | 140 | 80 |
| Consistency multiplier $\gamma$ | $4.25\lambda$ | $1.25\lambda$ | $4\lambda$ |
| Aggregate consistency penalty | $0.1\lambda$ | $0.2\lambda$ | $0.2\lambda$ |
| $\beta$-VAE weight | 1 | 1.3 | 2 |
| Predictor reg. ($\|w\|_2^2$) | 1 | 1 | 1 |
| Entropy reg. ($\mathbb{E}_{p_w(y|z)}[\log p_w(y|z)]$) | $0.5\lambda$ | $0.5\lambda$ | $0.5\lambda$ |
| Translation range ($\alpha^{(1)} = \alpha^{(2)}$) | $0.2 \times$ (image-width) | $0.2 \times$ (image-width) | $0.2 \times$ (image-width) |
| Rotation range ($\alpha^{(3)}$) | 0.4 rad | 0.5 rad | 0.4 rad |
| Shear range ($\alpha^{(4)}$) | 0.2 rad | 0.2 rad | 0.2 rad |
| Scale range ($\alpha^{(5)} = \alpha^{(6)}$) | 1.5 | 1.5 | 1.5 |
| Optimizer | ADAM | ADAM | ADAM |
| Learning rate | $3 \times 10^{-4}$ | $3 \times 10^{-4}$ | $3 \times 10^{-4}$ |

Table 3: Hyperparameter settings for semi-supervised learning experiments with our CPC-VAE.

## E    DATASET DETAILS

For each dataset considered in our paper, we provide a more detailed overview of its contents and properties.

### E.1    MNIST

**Overview.** We consider a 10-way exclusive categorization task for MNIST digits.

We use 28-by-28 pixel grayscale images.

**Public availability.** We will make code to extract our version available after publication.

**Data statistics.**    Statistics for MNIST are shown in Table 4.

| split | num. examples | label distribution |
|---|---|---|
| labeled train | 100 | [0.1 0.1 0.1 0.1 0.1 0.1 0.1 0.1 0.1 0.1] |
| unlabeled train | 49900 | [0.1 0.11 0.1 0.1 0.1 0.09 0.1 0.1 0.1 0.1] |
| labeled valid | 10000 | [0.1 0.11 0.1 0.1 0.1 0.09 0.1 0.1 0.1 0.1] |
| labeled test | 10000 | [0.1 0.11 0.1 0.1 0.1 0.09 0.1 0.1 0.1 0.1] |

Table 4: MNIST dataset.

### E.2    SVHN

**Overview.** We consider a 10-way exclusive categorization task for SVHN digits.

We use 32x32 pixel grayscale images.

**Public availability.** We will make code to extract our version available after publication.

**Data statistics.** Statistics for SVHN are shown in Table 5.

| split | num. examples | label distribution |
|---|---|---|
| labeled train | 1000 | [0.10 0.10 0.10 0.10 0.10 0.10 0.10 0.10 0.10 0.10] |
| unlabeled train | 62257 | [0.07 0.19 0.15 0.12 0.10 0.09 0.08 0.08 0.07 0.06] |
| labeled valid | 10000 | [0.07 0.19 0.14 0.12 0.10 0.09 0.08 0.08 0.07 0.06] |
| labeled test | 26032 | [0.07 0.20 0.16 0.11 0.10 0.09 0.08 0.08 0.06 0.06] |

Table 5: SVHN dataset.

### E.3 NORB

**Overview.**

We use 48x48 pixel grayscale images.

**Public availability.** We will make code to extract our version available after publication.

**Data statistics.** Statistics for NORB are shown in Table 6.

| split | num. examples | label distribution |
|---|---|---|
| labeled train | 1000 | [0.2 0.2 0.2 0.2 0.2] |
| unlabeled train | 21300 | [0.2 0.2 0.2 0.2 0.2] |
| labeled valid | 2000 | [0.2 0.2 0.2 0.2 0.2] |
| labeled test | 24300 | [0.2 0.2 0.2 0.2 0.2] |

Table 6: NORB dataset.

### E.4 CELEBA

**Overview.**

We use 64x64 pixel grayscale images. Images were cropped to square from the CelebA aligned variant and downscaled to our 64x64 resolution for computational efficiency. Labels were generated from the provided attributes. Our dataset used 4 classes: woman/neutral face, man/neutral face, woman/smiling, man/smiling.

**Public availability.** We will make code to extract our version available after publication.

**Data statistics.** Statistics for CelebA are shown in Table 7.

| split | num. examples | label distribution |
|---|---|---|
| labeled train | 1000 | [0.25 0.25 0.25 0.25] |
| unlabeled train | 21300 | [0.25 0.25 0.34 0.16] |
| labeled valid | 2000 | [0.25 0.25 0.25 0.25] |
| labeled test | 24300 | [0.27 0.23 0.35 0.15] |

Table 7: CelebA dataset.

