# OpenReview forum: "Learning Consistent Deep Generative Models from Sparse Data via Prediction Constraints"
_ICLR.cc/2021/Conference — Reject_

### Official Review · AnonReviewer4 · 2020-10-28
**Novel methodology for SSL-VAE, but too many loosely motivated heuristics**

**Rating:** 6
**Confidence:** 4

**Review:**

##########################################################################
### Summary
The paper presents a novel methodology for semi-supervised learning with variational autoencoders.
The discriminative part is added via prediction constraints to the standard VAE, resulting in PC-VAE.
The model is further extended with "consistency constraints" (an extra loss component that improves the prediction quality). The final model, CPC-VAE, is experimentally shown to perform better of similar to the competitors.

##########################################################################
### Reasons for score
 Overall, I give this paper a weak accept. The methodology is novel (to my knowledge), and is shown experimentally to outperform the competitors. However, there is no sufficient motivation behind the consistency constraints.
 The discussion of its effect is also lacking.

##########################################################################
### Pros
1) The paper is well written and is easy to understand.
2) A good introduction into the problem is given, with comprehensive background on unsupervised
and semi-supervised VAE.
3) The first proposed model, PC-VAE, is well motivated.
4) The experiments are thorough. Enough details are given (in the supplement) to ensure reproducibility.


##########################################################################
### Cons

1) Related work is scattered throughout the paper (intro, background, section 3.2), which makes
it hard to quickly place the work in perspective.
2) The hyperparemeter \lambda seems to play crucial role in balancing the generative and predictive parts. However,
the analysis of the relationship between \epsilon and \lambda is not thorough. There are no specific recommendations provided for the choice of \lambda based on the desired \epsilon.
3) The second model, CPC-VAE, is different from the PC-VAE in 2 ways: the consistency constraints  (eq 10-11)
and the aggregate label consistency constraint. Neither of these additions is motivated by the model or inference scheme and are heuristic regularisers. At the same time, the unmotivated heuristic regularizer used by [Kingma 2014] (and further works) was one of the big motivation points for this work.
4) The paper does not discuss reasons for the bad performance of the first proposed model (PC-VAE) in Table 2,
and only focuses on its modification (CPC-VAE). My intuition is that the discriminator part in PC-VAE overfits.
The issue seems to be fixed by the consistency constraints. This should be discussed and more intuition
should be provided as to how these consistency constraints fix the PC-VAE model.


##########################################################################
### Some typos/ minor comments

Fig. 2: The plot for the M2 model is unnecessary and can be misleading because the latent space
in M2 only provides auxiliary information to the class,
is not supposed to encode the class, and is not used to infer the class.

P.6 "[...], which forces the distribution of label predictions for unlabeled data is align with a known target distribution \pi."
Bad sentence/incorrect grammar. Rephrase, please, e.g. "[...], which forces the distribution of label predictions for
unlabeled data to be aligned with a known target distribution \pi."

---

> ### Author Response · Authors · 2020-11-25
> **Responses to reviewer 4**
>
> Thank you for your feedback!
>
> ***The hyperparameter \lambda seems to play crucial role in balancing the generative and predictive parts. However, the analysis of the relationship between \epsilon and \lambda is not thorough. There are no specific recommendations provided for the choice of \lambda based on the desired \epsilon.***
>
> As mentioned in another comment, we have revised with a new figure (see Fig. 10 in supplement) showing sensitivity to lambda and alpha for our MNIST model.
>
> Lagrange multiplier theory tells us that the relationship between \epsilon and \lambda is monotonic. Therefore, if an application-specific \epsilon needs to be achieved, a binary search can be used to find an appropriate \lambda value. If the goal is to achieve optimal test accuracy, as in our experiments, it is appropriate to treat \lambda as a hyperparameter and use validation data to evaluate a range of values. Empirically, we recommend starting \lambda inversely proportional to the fraction of labeled data. This gives the prediction loss a magnitude close to equivalent to the data being fully labeled. We find increasing \lambda from there gives good results as shown in the referenced figures.
>
> ***The second model, CPC-VAE, is different from the PC-VAE in 2 ways: the consistency constraints (eq 10-11) and the aggregate label consistency constraint. Neither of these additions is motivated by the model or inference scheme and are heuristic regularisers. At the same time, the unmotivated heuristic regularizer used by [Kingma 2014] (and further works) was one of the big motivation points for this work.***
>
> We have added additional detail to our manuscript to show how the consistency constraint is motivated directly from our modeling goals. Please see the decision network in figure 3 of the revised submission.  We emphasize that the consistency constraints are not simply a heuristic regularizer; they have a rigorous decision-theoretic motivation, just like the prediction constraints.
>
> ***The paper does not discuss reasons for the bad performance of the first proposed model (PC-VAE) in Table 2, and only focuses on its modification (CPC-VAE). My intuition is that the discriminator part in PC-VAE overfits. The issue seems to be fixed by the consistency constraints. This should be discussed and more intuition should be provided as to how these consistency constraints fix the PC-VAE model.***
>
> The semi-supervised performance of the PC-VAE is poor when compared to more advanced methods discussed in our work.  However PC-VAE still gives relative improvements over naive baselines like Supervised VAE and VAE-then-MLP, as shown by the new MNIST baseline in Table 2, and figures 1 and 2. If many or all examples are labeled, we find that PC-VAE is often an appropriate choice of model (see Figure 1 bottom row). However, our focus is when labels are rare.
>
> We believe figure 1 provides useful insight into the behavior of the PC-VAE model and why it often fails to predict well given few labeled examples. In particular note the top row PC-VAE subfigure (which gets 78.1% accuracy given 6 labeled examples). We see that this classifier cleanly separates all 6 labeled examples by class with simple, almost linear decision boundaries. This satisfies the prediction constraint and does *not* seem to overfit the labeled data. However, the learned decision boundaries do not account well for the underlying distribution of all data.
>
> In contrast, consider the corresponding CPC-VAE model in the top row of Figure 1. When we add a consistency constraint, for any observations near or on the decision boundary the sampled reconstructions will often fall on the opposite side of the boundary, leading to violations of the consistency constraint and a steep penalty incurred in the loss. Therefore the consistency constraint intuitively encourages that decision boundaries do not cut through densely populated parts of the unlabeled feature space, but rather go around them. This leads to far better generalization in practice as shown in our experiments. Thus, it is not overfitting that dooms PC-VAE, but an inability to inform its decision boundaries using unlabeled examples in regions far from labeled examples.
> We will revise the paper to include this intuitive understanding of why PC alone is not enough, but prediction and consistency constraints together work well.
>
> ***Fig. 2: The plot for the M2 model is unnecessary and can be misleading because the latent space in M2 only provides auxiliary information to the class, is not supposed to encode the class, and is not used to infer the class.***
>
> This is a fair criticism. We included it in the figure to highlight the interpretability of our latent space, which could make it useful for analyzing the model or as a method for visualization. We believed that the accuracy numbers shown made it clear that the M2 model was still working as intended. We are open to revising the figure if necessary.

---

### Official Review · AnonReviewer1 · 2020-10-28
**Good paper, with some concerns**

**Rating:** 5
**Confidence:** 4

**Review:**

The authors developed two constraints (and other techniques like using Spatial transformer, aggregate label consistency) that can better balance the generative and discriminative goals when learning VAEs and other generative models. The paper is easy to follow, the proposed methods look interesting and sound to me. According to Figure 1 and 2 in the paper, the proposed methods do have some benefits over other prior approaches. However, I have the below concerns which make me hesitate to accept this paper.

Regarding the overall performance:
One contribution, as highlighted in the abstract, is that the proposed method PC VAE and CPC VAE can boost semi-supervised classification performance. However, the major results shown in Table one and Table two seem do not strongly support this claim. In the Table one, the proposed approach only clearly outperforms other methods in NORB. Actually, many counterpart methods this paper compared with do not have results for NORB.
In the Table 2, clear benefits are from cycle-consistency rather than the predictive constraint, and the CPC also does not show clear benefits over M1+M2.

Regarding predictive constraints:
The PC VAE is very well motivated. However, it finally falls into a multitask objective (equation 9) for handling both labeled and unlabeled data. It is not clear that how PC itself seriously improve over prior VAE-based semi-supervised learning approaches.
In the Table two, PC does not outperform M2 and M1+M2. Also, in the Table two, can we compare PC directly with its deterministic counterpart?

Regarding consistent PC:
The proposed approach is a little bit similar to the ``"Unsupervised data augmentation for consistency training”. According the Table two, this technique improves the performance a lot. What if this consistency loss (and or other tricks like aggregate label consistency) was added to loss terms like (7)?

---

> ### Author Response · Authors · 2020-11-25
> **Responses to reviewer 1**
>
> Thank you for your feedback!
>
> ***Regarding the overall performance: One contribution, as highlighted in the abstract, is that the proposed method PC VAE and CPC VAE can boost semi-supervised classification performance. However, the major results shown in Table one and Table two seem do not strongly support this claim. In the Table one, the proposed approach only clearly outperforms other methods in NORB. Actually, many counterpart methods this paper compared with do not have results for NORB. In the Table 2, clear benefits are from cycle-consistency rather than the predictive constraint, and the CPC also does not show clear benefits over M1+M2.***
>
> Please see our top comments to R2 for discussion on why we believe our results on MNIST and SVHN are compelling. Most importantly, on SVHN, our method delivers accuracy that is as good as a state-of-the-art discriminative only method (VAT, Miyato et al 2017) which is not a generative model. Our method can deliver this performance and deliver good generative samples too.
>
> We have also added additional results on the CelebA dataset (see new Figure 6).  Our model outperforms discriminative baselines in a 4-way attribute classification task, while dramatically improving on unsupervised VAE models, and providing a generative model that can simulate labeled data. This dataset has been used with unsupervised VAEs in the past, such as in: “A General and Adaptive Robust Loss Function” (Barron, CVPR 2019).
>
>
> ***Regarding predictive constraints: The PC VAE is very well motivated. However, it finally falls into a multitask objective (equation 9) for handling both labeled and unlabeled data. It is not clear that how PC itself seriously improve over prior VAE-based semi-supervised learning approaches.***
>
> Our strong recommendation is to use the consistency-constrained model for semi-supervised learning tasks. As mentioned in other comments, the PC-VAE model still improves over naive methods (like the two stage unsupervised-VAE-then-MLP) and can be effectively used for fully-labeled data as shown in Figure 1 (though this is not the focus of our paper). We have expanded our manuscript with further discussion of the motivation for the consistency constraint as naturally arising from our principled generative modeling approach. See figure 3 in the revised submission for details.
>
> ***Regarding consistent PC: The proposed approach is a little bit similar to the ``"Unsupervised data augmentation for consistency training”. According the Table two, this technique improves the performance a lot. What if this consistency loss (and or other tricks like aggregate label consistency) was added to loss terms like (7)?***
>
> Thanks for suggesting the relevant related work on a method called Unsupervised data augmentation or UDA (Xie et al., NeurIPS 2020). We have added discussion of UDA in our revised paper in Sec. 3.2.
>
> UDA relies on highly-engineered data augmentation routines for specific domains (e.g. transforms specific to images like RandAugment or transforms for text like back-translation). In contrast, in our approach we learn a generative model to generate features x' that should have the same label as the original x. This makes our approach potentially more widely applicable to other domains where advanced augmentation routines are not easily available.
>
> The suggestion to add consistency regularization to other models (e.g. to M2's objective in our paper’s Eq. (7)) is interesting and likely would show some benefits, but this is beyond the scope of our work. We believe our ablation study in Table 2 (as well as earlier results in Fig. 1 and Fig 2.) adequately shows why M2 is not a preferred model. M2's predictions often get worse as generative capacity increases, while our approach does not have this weakness.

---

### Official Review · AnonReviewer3 · 2020-10-28
**Ok submission, experiments and results not very encouraging**

**Rating:** 6
**Confidence:** 4

**Review:**

The paper proposes a framework for semi-supervised settings to leverage both unlabeled data and (limited) labeled data where VAEs are trained subject to regularization terms from label information. More specifically, the proposed method trains a VAE and a NN classifier simultaneously by optimizing an objective that consists of the usual (unsupervised) variational lower bound, classification error for the labeled data based on the latent space, and consistency term for all data encouraging the same prediction for latent representations corresponding to the original and reconstructed version of a data point. The proposed method is compared with a few other deep generative semi-supervised learning methods on three image datasets.

+:
The paper is well written. It mentions some of the shortcomings of the previous approaches and motivates the proposed method and puts it into context. It is nice to see the details of the hyperparameter values used in the experiments. The results show reasonable performance in terms of classification accuracy, and demonstrate the effectiveness of using unlabeled data, predictive loss regularization, consistency loss regularization, and aggregate label consistency regularization.

-:
The proposed method involves several pieces that are put together. The regularization term from prediction loss, the regularization term from consistency loss, aggregate label consistency term, entropy regularization term for predictions, “noise-normal” likelihood, considering dimensions accounting for transformations in the latent space, KL scaling, …. The importance and contribution of some pieces to the performance is not clear, like the likelihood and entropy regularization. Especially that some like the likelihood, KL scaling, and transforms seem to be applicable to other VAE-based semi-supervised methods. Additionally, as all these terms have an associated multiplier, investigating sensitivity to these hyperparameters is important and will be helpful. The results in Table 1 and 2 are not very promising in terms of outperforming other existing methods.

Overall, the paper adds another VAE-based semi-supervised learning method that tries to address some of the limitations of previous approaches, but due to the above points, in my opinion in its current format is at a marginal level.

Post-rebuttal: Thanks for the authors’ response. After reading the responses and other comments and checking the updates to the paper, I retain the score and my recommendation at weak accept.

---

> ### Author Response · Authors · 2020-11-20
> **Responses to Reviewer 3**
>
> Thank you for your feedback and suggestions!
>
> ***The proposed method involves several pieces that are put together. The regularization term from prediction loss, the regularization term from consistency loss, aggregate label consistency term, entropy regularization term for predictions, “noise-normal” likelihood, considering dimensions accounting for transformations in the latent space, KL scaling, …. The importance and contribution of some pieces to the performance is not clear, like the likelihood and entropy regularization. Especially that some like the likelihood, KL scaling, and transforms seem to be applicable to other VAE-based semi-supervised methods.***
>
> KL scaling was included as a hyperparameter as it is common in other VAE works. L2 weight regularization and entropy regularization are similarly quite common for semi-supervised classifiers. Motivated by this past work, we allowed our hyperparameter optimization approach to include these parameters, but they appear to be mostly inessential. We have retested our best MNIST model without L2 and entropy regularization and setting the KL-scaling to 1. We left the other hyperparameters unchanged. Across 3 random trials we obtained test accuracy results of ~97% for each. As explained in another comment, spatial transforms are not necessarily applicable to the other models listed. We will revise our supplement to discuss this carefully.
>
> ***Additionally, as all these terms have an associated multiplier, investigating sensitivity to these hyperparameters is important and will be helpful***
>
> This is an excellent suggestion. We investigated the sensitivity to lambda (the PC strength) and gamma (the consistency strength) using our MNIST model, varying each hyperparameter while holding all others constant. We found that the sensitivity is fairly small across even orders of magnitude and behaves in line with our expectation. See section C of our revised supplement for the plots and details.
>
> ***The results in Table 1 and 2 are not very promising in terms of outperforming other existing methods***
>
> There are several reasons we feel that our results are strong despite not obtaining the highest accuracy on each dataset tested. Please see our comments to other reviewers for details.

---

### Official Review · AnonReviewer2 · 2020-10-29
**A New VAE model for semi-supervised problems**

**Rating:** 5
**Confidence:** 4

**Review:**

Summary:

This paper proposes a new VAE framework for semi-supervised problems, which uses the latent representation \\(z\\) to reconstruct input image \\(x\\) and to serve as the features for the classification of the label of \\(x\\). Based on this framework, the paper also proposes additional "cycle" losses, where the label prediction based on \\(\overline{z}\\) of \\(\overline{x}\\) is close to the true label (for data with supervisions) or the label of \\(x\\) (for data without supervisions). In addition, the paper also introduces another loss term of "aggregate label consistency" and applies other techniques including noise likelihood and STN. The proposed approach outperforms M1 and M2 of Kingma et al. 2014 on the synthetic dataset and shows more stability than M1 and M2. It seems that the performance advantage of the proposed method over others is not very significant on real datasets.

Pros:
- The idea of using  \\(z\\) to reconstruct input image \\(x\\) and to serve as the features for the classification of the label of \\(x\\) is simple and straightforward (in a good way). The consistent PC loss is also intuitive and adds more credits to the technical depth of the paper.

- The released code is a plus of the reproducibility of the paper.

- The writing of the paper is clear in general.

Cons:

- The demonstration of the motivations of the paper is a bit confusing to me. The half-moon data in Fig 1 is simple, therefore C=2 is good enough for the models to fit. The proposed model is more stable when C increases. But it's less motivated to increase C in this case given the data simplicity. So C=14 might not be a good setting in this case. Also in Fig 2, MNIST is clearly more complex than the half-moon data. Therefore, C=2 might not be a good setting in this case. The settings of Fig 1 and 2 are less intuitive to demonstrate the motivations.

- One of the main claims is the high running cost of M2. The basic framework (PC) of the paper might have an advantage in running cost. But I'm wondering if it still has when added with the consistent PC loss, aggregate label consistency, and also STN. With those components, the complexity of the proposed model clearly increases and it seems that without those components the proposed model has no clear advantage over others. The paper didn't provide any empirical comparison of the running cost.

- There seems to be no clear advantage of the proposed method over others in terms of performance. There are wins and loses across different datasets. I also have a concern about the fairness of the comparison. For example, it is reasonable to expect that with STN, the performance of M1/M2/others might be also improved. In the comparison, CPC is with STN but the numbers of M1/M2 are from their original papers. Therefore, it is unclear where the performance gain comes from.

-  As the proposed method involves several components. Some of them are not detailed enough. For example, it's a bit unclear to me how the noise likelihood is employed given a short paragraph of description.

-------------------------------------------------------------------------------------------------------------------------------------------------------------------------------

The author response addresses some of my concerns. So I have updated my rating from 4 to 5. I recognise the novelty of the proposed framework but I feel that the main framework has not shown a clear performance advantage given several other components are added. Therefore, I am unable to give a clear recommendation for acceptance.

---

> ### Author Response · Authors · 2020-11-20
> **Responses to Reviewer 2**
>
> Thank you for your helpful feedback!
>
> ***The demonstration of the motivations of the paper is a bit confusing to me. The half-moon data in Fig 1 is simple, therefore C=2 is good enough for the models to fit. The proposed model is more stable when C increases. But it's less motivated to increase C in this case given the data simplicity. So C=14 might not be a good setting in this case. Also in Fig 2, MNIST is clearly more complex than the half-moon data. Therefore, C=2 might not be a good setting in this case. The settings of Fig 1 and 2 are less intuitive to demonstrate the motivations.***
>
> Our goal across Figure 1 and FIgure 2 is to assess how sensitive different methods are to choices like the encoding size C. We should prefer a method that delivers good performance for a wide range of values, instead of expecting practitioners to do an expensive grid search to find exactly the right value for each new dataset. Across these figures as well as our ablation study on MNIST in Table 2, we show that one is unlikely to be able to improve performance of M2 by increasing its model capacity (either encoding size or the depth of the decoder network). Instead, we see a strong opposite effect: adding capacity makes M2’s predictions worse. This could lead to considerable difficulty in applying that model in practice.
>
> For Figure 1 (half moon), yes this is a simple dataset where M2 with C=2 performs well. But a modest increase in C leads to noticeably worse performance for M2, while our method is stable.
>
> For Figure 2 (MNIST visualization with C=2), yes it is true that compared to C=2, larger C values are better for MNIST, as we show later in Table 2, where C=50 performs much better (88% accuracy) as in the original results reported for the M2 model. We use C=2 in Figure 2 so that we can effectively visualize the encoded data, allowing us to highlight the intuitive differences between the models and demonstrate the failure modes of our baselines. We believe the interpretability of the latent space is also a benefit of our approach (in contrast with M2).
>
> Overall, we intend these results to show that M2 is not the right model for achieving both discriminative and generative goals, as adding generative capacity seems to often harm its prediction quality. This provides motivation for our prediction constrained approach.
>
>
>
>
> ***One of the main claims is the high running cost of M2. The basic framework (PC) of the paper might have an advantage in running cost. But I'm wondering if it still has when added with the consistent PC loss, aggregate label consistency, and also STN. With those components, the complexity of the proposed model clearly increases and it seems that without those components the proposed model has no clear advantage over others. The paper didn't provide any empirical comparison of the running cost.***
>
> Thank you for the suggestion for an empirical runtime cost comparison. Using our MNIST models as reported in table 2, we found the PC-VAE took on average 38ms per training step and the CPC-VAE averaged 84ms. In contrast, the M2 model required 254 ms per training step. Thus, we find our CPC-VAE even with all extra components (aggregate loss, STN) is about 3 times faster than M2, which we believe validates our claims of significant speedup.
>
> We intend to add these results to the final draft of our paper, as well as the asymptotic analysis below. Naturally, these results will of course be affected by the size of the networks used, but we did not find it necessary to use larger networks for our model in practice.
>
> For our proposed CPC method, the encoder and decoder networks are the most expensive functions to compute by at least an order of magnitude. The aggregate label consistency is simply an additional loss computed using the existing predictions, so its runtime cost is negligible. For the spatial transformations, we are using a spatial transformer layer, but not a separate spatial transformer network. We simply repurpose some of the latent dimensions to encode transformations explicitly. Therefore the additional cost is only that of the transformer layer itself, which is comparable to a single additional convolutional layer.
>
> M2 is significantly slower because it must run both the encoder, decoder and prediction networks once for each class in order to compute the loss.  In contrast, the consistency constrained model can compute its loss running the encoder/decoder networks twice no matter how many classes are considered (once for the original example, once for the generated example that should be consistent with the original). In other words, we argue that the asymptotic runtime complexity of a training step is significantly better for our method as it does not depend on the number of classes. As expected this is borne out in practice in the numbers above, though the scaling is not exact, likely due to data loading bottlenecks.

---

> > ### Author Response · Authors · 2020-11-20
> > **Responses to Reviewer 2 (cont.)**
> >
> > ***There seems to be no clear advantage of the proposed method over others in terms of performance. There are wins and loses across different datasets.***
> >
> > In Table 1, we emphasize that our CPC-VAE beats all other semi-supervised VAE methods on both SVHN and NORB. We suggest that good performance on these datasets is more relevant than the over-studied MNIST dataset.
> >
> > Focusing on Table 1 MNIST results, the ADGM/SDGM models reported here that achieve best accuracy use a more complex model with significantly more parameters than M2 or our method. In contrast, for our MNIST result we chose our encoder and decoder network to match the ones used by the original M2 model for fairness. Using the same hyperparameter settings and replacing our network with small resnet (equaivalent to a WRN-20-1, using the wide resnet notion) our performance improves to ~98.5%, very close to the SDGM’s 98.7%. We suspect with further optimization, we could meet or exceed the ADGM results.
> >
> > On SVHN, we do agree that a non-generative baseline (VAT) achieves similar performance to our method. However, it is worth noting that VAT is not VAE-based, nor is it a generative model. Our goal is to perform good semi-supervised learning while retaining the useful properties of a generative VAE: interpretability, image generation and the potential for auxiliary tasks using the same model, such as inpainting or alignment. VAT can do none of these things. We can match its performance while retaining all of these properties.
> >
> > ***I also have a concern about the fairness of the comparison. For example, it is reasonable to expect that with STN, the performance of M1/M2/others might be also improved. In the comparison, CPC is with STN but the numbers of M1/M2 are from their original papers. Therefore, it is unclear where the performance gain comes from.***
> >
> > As part of our ablation study in Table 2, we include results on MNIST without transformations and demonstrate that we still outperform M2 by a significant margin (91.93% vs 83.32% with same 2 layer architecture). We argue that this ablation clearly signals that the relative benefit is due to our improved training objective. To ensure fairness to the M2 baseline,  we verified that the performance of our M2 reimplementation matched the reported performance in the original paper closely and we tested a range of hyperparameters.
> >
> > The spatial transformation could not easily be applied to the M1 + M2 model in the same way. In that setup the M1 and M2 models are trained in two stages and only the unsupervised M1 portion operates on 2 dimensional image features. The spatial transformation could be applied to the M1 portion of the model, but its training would not interact with the labels as that is trained in an unsupervised fashion. Furthermore in the SVHN and NORB results reported for the M1 + M2 model, the authors state that their method worked best on a PCA decomposition of the data. In that regime the spatial structure is lost and a transformer layer would be inapplicable.
> >
> >
> > ***As the proposed method involves several components. Some of them are not detailed enough. For example, it's a bit unclear to me how the noise likelihood is employed given a short paragraph of description.***
> >
> > We have added additional detail on the Noise likelihood to the main text describing its density function and its use as a likelihood in our PC-VAE model. We are happy to further address any additional lack of clarity in our methodology if needed.

---

### Author Response · Authors · 2020-11-25
**Updates to our submission**

Thank you again to all the reviewers for your helpful feedback and comments! We have uploaded a revised version of our submission with changes to address several of the common concerns:
- A new figure (Figure 6) has been introduced to show classification results for our method and baselines on an SSL variant of the CelebA dataset
- A new figure (Figure 3) has been introduced to provide further detail on the consistency constraint and its motivation as a natural way to achieve our generative modeling goals.
- Our ablation study in Table 2 has been expanded to include results with 2-stage training (discriminative classifier trained on unsupervised VAE encodings) and with a variant of the M2 model using the “Noise-Normal” likelihood.
- We updated Table 1 to include our best MNIST result using a WideResNet (WRN), not just an MLP as before.  This increases the accuracy of the CPC-VAE to 98.3%.
- Section 3.2 includes updated discussion on the UDA Method (Xie et al. 2020).
- Section 3.3 has been expanded to clarify the definition and use of the “Noise-Normal” likelihood.


We emphasize that our contribution is a new way to perform semi-supervised learning for deep generative models that achieves both discriminative and generative goals. Our method is competitive with state-of-the-art discriminative-only SSL methods like VAT (Miyato et al) on classification tasks, but can also deliver a generative model with compelling visual samples. Previous alternatives like M2 that can do SSL with a generative model have worse classifier performance than our CPC-VAE, worse runtime performance (see our runtime discussion response to Reviewer 2), and bad properties like classifier deterioration as the generative model capacity gets larger.

---

### Decision · Program_Chairs · 2021-01-07
**Final Decision**

**Decision:**

Reject

**Comment:**

This paper presents a semi-supervised model (named CPC-VAE) that trains a variational autoencoder (VAE) and a NN classifier simultaneously. The method maximizes an ELBO subject to a task-specific prediction constraint and a consistency constraint. The constraints are defined as some expectations of the variational posteriors. Such constraints are known as posterior regularization. Though the consistency constraint seems to be new, the prediction constraint has been well-examined under deep generative models (see e.g., max-margin deep generative models for (semi-)supervised learning, IEEE TPAMI, 2018). The paper needs more a thorough analysis and comparison.